# Mechanistic Insights into the Anti-Proliferative Action of Gut Microbial Metabolites against Breast Adenocarcinoma Cells

**DOI:** 10.3390/ijms242015053

**Published:** 2023-10-10

**Authors:** Kayla Jaye, Muhammad A. Alsherbiny, Dennis Chang, Chun-Guang Li, Deep Jyoti Bhuyan

**Affiliations:** 1NICM Health Research Institute, Western Sydney University, Penrith, NSW 2751, Australia; 19255718@student.westernsydney.edu.au (K.J.); muhammad.alsherbiny@pharma.cu.edu.eg (M.A.A.); d.chang@westernsydney.edu.au (D.C.); c.li@westernsydney.edu.au (C.-G.L.); 2Pharmacognosy Department, Faculty of Pharmacy, Cairo University, Cairo 11562, Egypt; 3Innovation Centre, Victor Chang Cardiac Research Institute, Sydney, NSW 2010, Australia; 4School of Science, Western Sydney University, Penrith, NSW 2751, Australia

**Keywords:** gut microbial metabolites, postbiotics, breast cancer, gut microbiome, sodium butyrate, inosine, nisin

## Abstract

The gut microbiota undergoes metabolic processes to produce by-products (gut metabolites), which play a vital role in the overall maintenance of health and prevention of disease within the body. However, the use of gut metabolites as anticancer agents and their molecular mechanisms of action are largely unknown. Therefore, this study evaluated the anti-proliferative effects of three key gut microbial metabolites—sodium butyrate, inosine, and nisin, against MCF7 and MDA-MB-231 breast adenocarcinoma cell lines. To determine the potential mechanistic action of these gut metabolites, flow cytometric assessments of apoptotic potential, reactive oxygen species (ROS) production measurements and proteomics analyses were performed. Sodium butyrate exhibited promising cytotoxicity, with IC_50_ values of 5.23 mM and 5.06 mM against MCF7 and MDA-MB-231 cells, respectively. All three metabolites were found to induce apoptotic cell death and inhibit the production of ROS in both cell lines. Nisin and inosine indicated a potential activation of cell cycle processes. Sodium butyrate indicated the possible initiation of signal transduction processes and cellular responses to stimuli. Further investigations are necessary to ascertain the effective therapeutic dose of these metabolites, and future research on patient-derived tumour spheroids will provide insights into the potential use of these gut metabolites in cancer therapy.

## 1. Introduction

The role of gut microbiota in the treatment of breast cancer is a relatively new area of oncology research. Several metabolites (also known as postbiotics) produced by the gut microbiota have shown potential anti-proliferative activity against cancer cells as well as synergies with standard anticancer drugs by enhancing their activity and reducing side effects [1]. Short-chain fatty acids (SCFAs), produced by different bacterial species, are the most abundant type of metabolites within the human gut, with butyrate being the most common SCFA produced by the *Firmicutes* [1]. This group of metabolites are derived from the breakdown of dietary fibre, in which fermentation of non-digestible carbohydrates results in the production of SCFAs [2]. Nisin is produced via bacterial fermentation of food products, such as yoghurt by the Gram-positive *Lactobacillus lactis*, and is the most produced bacteriocin in the human gut [1]. Under normal physiological conditions, nisin limits colonisation by Gram-positive bacterial species within the gut in a phenomenon known as ‘colonisation resistance’, and triggers changes in the cellular membrane potential of the entire cell [1]. Inosine is produced via the metabolic conversion of adenosine by the adenosine deaminase enzyme [1] and also by certain gut microbial species including *Bifidobacterium pseudolongum* [3]. Sodium butyrate is the sodium salt of butyrate and was selected for the present study instead of butyric acid as the latter, when administered orally, is unable to reach the lower gastrointestinal tract (GIT) due to its absorption in the upper GIT [4]. Butyrate also possesses unfavourable pharmacological properties, including a short half-life and multigram doses required to achieve therapeutic concentrations in animal models [5]. The molecular mechanisms of action of these gut microbial metabolites have not been investigated in depth against breast cancer cells.

The MCF7 human breast adenocarcinoma cell line is one of the most utilised breast cancer cells in cancer research. The MCF7 cell line is highly invasive and dependent on oestrogen and progesterone hormones and expresses substantial levels of the oestrogen receptor (ER) alpha, which is clinically representative of common invasive breast cancers that characteristically express ER [6]. Additionally, the MCF7 human breast cancer cell line is representative of the in vivo mammary gland structure, due to its ability to form directionally oriented microtissues containing a luminal space when grown as a 3D cell culture model [7]. Due to these factors, the MCF7 cell line has been used extensively in breast cancer-related studies and is ideal for evaluating the activity of potential antiproliferative activity of drugs. The application of this cell line in preclinical breast cancer research has been instrumental in determining new potential anticancer agents and contributing to improvements in patient outcomes [6]. Triple-negative breast cancers (TNBCs) account for approximately 15 to 20% of all breast cancer cases and have a poor prognosis, which warrants investigation to define targeted therapies [8]. The MDA-MB-231 is a TNBC cell line, representing a highly aggressive form of breast cancer with limited treatment options and low survival rates [9]. The MDA-MB-231 cells are oestrogen-independent and demonstrate representative epithelial to mesenchymal transition (EMT) associated with metastatic breast cancer progression [9]. It is also characterised by the non-expression of progesterone receptors and the lack of overexpression of HER2, and typically associated with *BRCA1/2* germline mutations that hereditarily predispose young women to breast cancer [8]. This cell line is used less commonly in in vitro research compared to the MCF7 breast cancer cell line; however, it is a preferred cell line for use in xenograft animal models [10]. In immune-compromised mice, these breast cancer cells present with preferential growth in the mammary fat pad of this animal model and initiate spontaneous metastasis within the host lymph nodes that are derived from primary tumours [10]. The action of MDA-MB-231 cells in xenograft animal models is clinically representative of the mechanism in which these cancerous cells act in the human host, due to the coordination of EMT in breast cancer metastasis [9]. To determine the potential cytotoxicity of the selected gut metabolites against healthy breast tissue, the metabolites were also tested against the normal breast epithelial MCF10A cell line.

This study was designed to gain a mechanistic understanding of sodium butyrate, nisin and inosine against the oestrogen-dependent MCF7 and triple negative MDA-MB-231 human breast adenocarcinoma cell lines using cellular, molecular and proteomics assays.

## 2. Results and Discussion

### 2.1. Anti-Proliferative Activity of the Three Gut Microbial Metabolites against the MCF7 and MDA-MB-231 Human Breast Adenocarcinoma Cells

The cell viability of the MCF7 cells following a 72-h treatment with different concentrations of sodium butyrate, nisin and inosine was evaluated using the Alamar Blue assay. This assay determined that sodium butyrate exhibited the most substantial anti-proliferative activity in a dose-dependent manner (*p* < 0.0001), followed by inosine and nisin (Table 1). As observed with the MCF7 cell line, the most effective metabolite in inhibiting MDA-MB-231 cell viability was sodium butyrate, followed by inosine (*p* < 0.0001) and nisin (*p* < 0.0001) (Table 1). All three metabolites demonstrated dose-dependent cell growth inhibition in the range of 0.03125–3 mg/mL.

At 3 mg/mL (27.25 mM) and 2 mg/mL (18.17 mM), sodium butyrate exhibited MCF7 anti-proliferative activity of 81.73 ± 4.81% and 70.86 ± 7.24%, respectively. Previously, sodium butyrate at 2.5 mM was found to inhibit the growth of the MCF7 cells by 85–90% [11]. Another in vitro study previously observed only 34% growth inhibition of the MCF7 cells with 1 mM of butyrate [12]. In our study, the IC_50_ value of sodium butyrate against the MCF7 cells was found to be 0.576 mg/mL (5.23 mM) using the Alamar Blue assay (Table 1), with an R^2^ value of 0.9513, which is demonstrative of the reliability of the obtained results. Based on the R^2^ value, the findings of the cell viability study for sodium butyrate were statistically relevant, as this value was above the confidence interval (CI) of 95%. Previous preclinical research has also ascertained the IC_50_ value of sodium butyrate against the MCF7 breast cancer cell line with the following values: 1.26 mM [13] and 2.5 mM after 24 h [14], and 0.5 mM after 9 days exposure [11]. Earlier studies on the anticancer potential of gut metabolites have primarily utilised the MTT assay; however, the present study used the Alamar Blue assay to understand the anti-proliferative effects of these metabolites. The Alamar Blue assay is non-toxic and does not require lysed cells, unlike the MTT assay, and has been found to demonstrate more sensitivity towards certain compounds compared to MTT [15]. Similar to the MCF7 cell line, sodium butyrate presented with a mean inhibition of 80.8% at 3 mg/mL (R^2^ = 0.9315; IC_50_ = 0.557 mg/mL or 5.06 mM) against the MDA-MB-231 cells. Sodium butyrate has been assessed against the MDA-MB-231 cells previously, but to a lesser extent than MCF7 cells. Nonetheless, the anti-proliferative activity of sodium butyrate in the present study correlated with previous findings on the MDA-MB-231 cells [11,14]. The inhibitory percentage values were dose-dependent, where doses more than 0.5 mg/mL showed >50% cell growth inhibition and less than 0.25 mg/mL displayed <50% cell growth inhibition. Previously, sodium butyrate exhibited 85–90% inhibition against the MDA-MB-231 cells at 2.5 mM (0.275 mg/mL) with an IC_50_ value of 0.6 mM (0.066 mg/mL) [11]. That study utilised the Burton assay to measure the antiproliferative effect through variation of DNA content [11], whereas the Alamar Blue assay used in our study measures the metabolic activity of the cells. This could account for the discrepancies in activity observed between that study and the present study. Another factor of consideration in the assessment of butyrate as an anticancer agent is the ‘butyrate paradox’ phenomenon, in which low concentrations of butyrate can be carcinogenic, whereas higher concentrations of butyrate have anticancer effects [16]. Whilst the sodium salt of butyrate has not been explicitly studied clinically, SCFAs have been investigated in clinical settings against colorectal and breast cancer, as well as solid cancer tumour types [17,18,19]. Specifically, butyrate and propionate have been assessed clinically in the context of breast cancer, in which it was observed that patients with premenopausal breast cancer presented with a reduction in SCFA-producing bacteria and key SCFA-producing enzymes compared to healthy premenopausal women, which could serve as a diagnostic tool in premenopausal breast cancer [17]. As sodium butyrate is considered a more stable form of butyrate [4], further studies evaluating its efficacy in preclinical settings are necessary before clinical investigation.

Inosine demonstrated the growth inhibition value of 47.78% ± 8.89 at 3 mg/mL, respectively (R^2^ = 0.8027), with an IC_50_ value of 2.743 mg/mL (10.23 mM) against the MCF7 cells (Table 1). Whilst the cell inhibition percentages were significantly lower than sodium butyrate (*p* < 0.05), this study is amongst the first to assess the anti-proliferative activity of inosine against the MCF7 breast cancer cells and serves as a foundation for future research regarding its use as a natural anticancer agent. A previous study assessed inosine against the MCF7 and MDA-MB-231 breast cancer cells; however, the purpose of the study was to explore its cytoprotective role in breast cancer hypoxic conditions, which is different from the scope of the current study [20]. The authors reported that inosine, not adenosine, was the primary cytoprotective compound in breast cancer hypoxic conditions [20]. Inosine was the second most effective gut microbial metabolite in inhibiting the growth of the MDA-MB-231 cells dose-dependently (Table 1) in this study. At 3 and 2 mg/mL, inosine initiated 48.85% and 36.09% mean cell growth inhibition, respectively, with an IC_50_ value of 3.262 mg/mL (12.16 mM). The R^2^ value determined in the cell inhibition effect of inosine against the MDA-MB-231 cells was 0.9485, indicating that the data fit the regression model in the goodness-of-fit test. As with the MCF7 cell line, inosine has been assessed against the MDA-MB-231 cell line previously; however, the purpose of that study was to determine the role of inosine in breast cancer hypoxic conditions [20]. Therefore, our study is amongst the first to assess the anti-proliferative potential of inosine against the MDA-MB-231 cell line and will provide a framework for future investigation on this gut microbial metabolite and its bioactivity. Similar to sodium butyrate, inosine has not been examined in clinical settings.

Nisin has been primarily investigated against colorectal cancer cell lines; however, a few in vitro studies have previously observed high cytotoxicity and inhibition of cell proliferation against the MCF7 cells [21,22]. Nisin presented with cell inhibition values of 14.34% and 13.31% at 3 mg/mL (894.44 μM) and 2 mg/mL (596.29 μM), respectively (Table 1), against MCF7 cells in the present study. These findings contrasted with previous in vitro studies, which reported that nisin exhibited high cytotoxicity and selectivity against the MCF7 cells in a concentration-dependent manner using the MTT assay, but showed broad IC_50_ values of 5 μM (0.0168 mg/mL) [21] and 105.46 μM (0.354 mg/mL) [22]. The significant difference in the activity of nisin observed between our study and the literature could be attributed to the fact that the two prior studies utilised the MTT assay to assess the cell viability of nisin, whereas in the present study, the Alamar Blue assay was used. Similar to sodium butyrate and inosine, nisin has not been specifically studied clinically as an anticancer agent. However, other bacteriocins including the azurin-p28 peptide have been assessed against the central nervous system (CNS) tumours and p53(+) metastatic solid tumours clinically [23,24]. The potential anti-proliferative activity of nisin against the MDA-MB-231 breast cancer cells was not investigated before. In the present study, nisin presented with significantly lower (*p* < 0.05) cell growth inhibition activity compared to sodium butyrate and inosine at the 1–3 mg/mL range against the MDA-MB-231 cells (Table 1). At its highest tested concentration of 3 mg/mL (894.44 μM), the mean cell growth inhibition of nisin against the MDA-MB-231 cells was 24.46%. The anti-proliferative activity of nisin against the MDA-MB-231 cells was similar to that of the MCF7 cells.

### 2.2. Cytotoxicity of Gut Metabolites against Normal MCF10A Human Breast Epithelial Cells

The cytotoxicity of drugs to normal breast tissue due to the lack of selectivity is a serious concern in anticancer drug discovery. In addition to determining the cytotoxicity of sodium butyrate, nisin and inosine against the MCF7 and MDA-MB-231 breast cancer cells, their potential cytotoxicity against the normal MCF10A human breast epithelial cells was also evaluated using the Alamar Blue assay (Table 1). At the highest tested concentration of 3 mg/mL, nisin, inosine, and sodium butyrate demonstrated cytotoxicity against the MCF10A breast epithelial cells, with cell growth inhibition values of 69.43%, 87.64%, and 89.23%, respectively. Nisin, inosine, and sodium butyrate displayed IC_50_ values of 1.63 mg/mL (485.98 μM; R^2^ < 0.5), 0.655 mg/mL (2.44 mM; R^2^ = 0.7489), and 0.655 mg/mL (5.95 mM; R^2^ = 0.8740), respectively, against the MCF10A cells. The IC_50_ values of nisin and inosine against the MCF10A cells were lower than those for the MCF7 cells. These findings are contradictory to a previous in vitro study which identified that nisin expressed selective cytotoxicity for the MCF7 cells [21]. Comparatively, the IC_50_ value of sodium butyrate against the MCF10A cells (0.655 mg/mL; 5.95 mM) was greater than that of the MCF7 (0.576 mg/mL; 5.23 mM) and MDA-MB-231 (0.557 mg/mL; 5.06 mM) cells. This indicated that sodium butyrate had a greater cytotoxicity against the breast cancer cells than the normal breast epithelial cells. These findings warrant further investigation, as limited research has been conducted regarding the potential cytotoxicity of gut microbial metabolites to normal breast tissue. Compared to the current 2D cell assay, these gut metabolites may present differently in 3D cell culture or animal models, as they are more clinically representative. The appropriate administration route to avoid cytotoxicity to normal cells is another important consideration. Overall, future studies will need to determine the potential risk of gut metabolite administration to normal breast tissue and approaches to avoid cytotoxicity.

### 2.3. Inhibition of ROS Production by the Gut Metabolites in the MCF7 and MDA-MB-231 Cells

The increased production of ROS within a host can lead to carcinogenesis due to the initiation of DNA damage in normal cells and consequent apoptosis, whereas lower concentrations of ROS contribute to signalling processes involved in maintaining physiological homeostasis [25]. Previous studies have shown that the MCF7 cells produce high basal levels of intracellular ROS [26]. Therefore, identifying whether nisin, inosine, and sodium butyrate contribute to an increase in production of ROS was fundamental to assess a potential aggravating risk factor to the progression of breast cancer. Overall, the three gut metabolites were found to inhibit the production of ROS in the MCF7 cells in comparison to the negative control after 4 h of treatment (Figure 1). The positive control doxorubicin (DOX; 1 μM) increased the production of ROS by 11.22% compared to the negative control. The inhibitory effect of nisin on ROS production was found to be 61.14% and 59.63% at 3 and 2 mg/mL, respectively, compared to the negative control. Inosine demonstrated ROS inhibition values of 65.18% and 58.52% at 3 and 2 mg/mL, respectively, whereas sodium butyrate showed ROS inhibition values of 43.12% (3 mg/mL) and 49.16% (2 mg/mL) compared to the negative control. At 3 mg/mL, inosine exhibited significantly more ROS inhibition than did sodium butyrate (*p* = 0.0126). Previously, sodium butyrate was observed to increase levels of ROS after 48 h at 1, 5, and 10 mM, which initiated the induction of apoptosis in the MCF7 cells [27]. Our study assessed the impact of higher concentrations of sodium butyrate (27.25 and 18.17 mM) on ROS production by the MCF7 cells after 24 h of treatment compared to the previous study. Another study also observed that inosine increased the intracellular ROS levels in the Caco-2 colon cancer cells after 3 h at various concentrations [28].

This study also assessed whether the three gut microbial metabolites increased or decreased the abundance of ROS within the MDA-MB-231 breast cancer cell line after 4 h of treatment. Compared to the negative control (supplementary buffer), all three gut metabolites inhibited the production of ROS in the MDA-MB-231 cells (Figure 1). Nisin inhibited the production of ROS by 60.97% and 61.47% at 3 mg/mL (894.44 μM) and 2 mg/mL (596.29 μM), respectively. Inosine showed statistically similar (*p* > 0.05) ROS inhibition values of 62.94% and 65.03% at the two highest concentrations of 3 (11.18 mM) and 2 (7.46 mM) mg/mL, respectively. Compared to inosine at 3 and 2 mg/mL, sodium butyrate at 3 mg/mL (27.25 mM) had a significantly lower (*p* < 0.05; Figure 1) ROS inhibitory effect (50.37% inhibition). These findings are consistent with the MCF7 anti-proliferative data. The impact of sodium butyrate on the ROS levels of the MDA-MB-231 cells has not been assessed to date. The effect of DOX (1 μM) on the production of ROS was assessed as the positive control in this study and was found to increase the production of ROS by 22.58% in the MDA-MB-231 cells compared to the negative control.

### 2.4. Flow Cytometric Analysis of Apoptosis in the MCF7 and MDA-MB-231 Cells Using Annexin V-CF Blue and 7-AAD

Apoptosis is a form of programmed cell death that, amongst numerous biological properties, functions to prevent cancer in the removal of cancerous cells from the host [29]. Given the significant role apoptotic cell death plays in cancer inhibition, targeting this molecular action is a primary aim of many novel anticancer agents. To ascertain whether the three gut microbial metabolites did initiate apoptosis in the MCF7 and MDA-MB-231 breast cancer cells after 24 h of treatment, the annexin V-CF blue and 7-AAD flow cytometric assay was utilized (Figure 2).

Of the three gut microbial metabolites, sodium butyrate treatment (3 mg/mL; 27.25 mM) after 24 h exhibited significantly more (*p* < 0.001; Figure 2a,b) early- and late-stage apoptotic cell death compared to the negative control, with a combined apoptotic initiation of 28.95% against the MCF7 cells. Previous studies identified that sodium butyrate induced cell cycle arrest in the G_2_ or G_2_/M phases against MCF7 breast cancer cells [11,14,30]. These studies on MCF7 cells reported that the mechanisms involved in cell cycle arrest included the inhibition of cell growth in a p53-independent manner and induction of apoptotic cell death via the Fas/Fas L system [14], and the inhibition of histone H1 kinase activity with increased levels of p21^waft^ [30]. Another study identified that sodium butyrate induced apoptosis dose-dependently, i.e., a higher concentration (5 mM) led to a higher apoptotic induction [13]. Sodium butyrate was also found to initiate apoptotic cell death via cell cycle arrest at the G_1_/G_2_ phase [13,31]. Other studies have found sodium butyrate induces apoptosis through increased expression of 15-lipoxygenase-1 (15-Lox-1) [32], decreased intracellular glutathione levels [33], increased p21^waf1/cip1^ levels and interactions with PCNA [34], increased expression of death receptors [35], increased levels of Fas signalling via the Fas/Fas L system independently of p53 [14], and the induction of cell cycle arrest in the G_2_/M growth phase [11]. Morphological changes in the MCF7 cells when treated with sodium butyrate indicative of apoptosis were also reported previously, which was further validated by the observation that levels of the Bcl-2 protein were reduced by higher concentrations of sodium butyrate, correlating with the increased levels of apoptosis [36].

In the present study, sodium butyrate displayed the combined early- and late-stage apoptotic MDA-MB-231 cell percentage of 36.93% after 24 h treatment at 3 mg/mL (27.25 mM) (Figure 2d), which correlated with the anti-proliferative activity data (80.8% cell growth inhibition at 27.25 mM after 72 h). Furthermore, sodium butyrate exhibited significantly more early- and late-stage apoptotic MDA-MB-231 cells compared to the negative control (*p* < 0.0001), nisin (*p* = 0.0024), and inosine (*p* = 0.0007). In contrast to these findings, previous studies did not observe any induction of apoptosis by sodium butyrate in the MDA-MB-231 cells, although the MCF7 cells underwent apoptosis after sodium butyrate treatment of 2.5 mM for 48 h [14]. Interestingly, although the authors did not observe apoptosis induction in the MDA-MB-231 cells, sodium butyrate was found to induce cell cycle arrest in the G_2_/M phase [14]. Likewise, a study by Lallemand et al. (1999) acknowledged that sodium butyrate induced cell cycle arrest in the G_2_/M phase [30]. Another in vitro study was also unable to detect the initiation of apoptosis by sodium butyrate at 2.5 mM after 9 days using the gel electrophoresis and flow cytometric analyses in the oestrogen-independent cell lines including the MDA-MB-231 cells [11]. These findings may be ascribed to the butyrate paradox, in which lower concentrations of sodium butyrate show carcinogenic effects, whereas higher concentrations are required to demonstrate anticancer effects. The discrepancies observed between our study and previous reports could be associated with the high concentrations of sodium butyrate and treatment time point used in our study. In addition to these studies, other in vitro studies assessing the activity of sodium butyrate against the MDA-MB-231 cell line did not investigate its apoptotic induction potential [30,37]. Interestingly, the initiation of cell cycle arrest in the G_2_/M growth phase leading to the induction of apoptotic cell death in pyruvate carboxylase-knockdown MDA-MB-231 cells was observed previously [38]. As such, the present study may be the first to investigate the induction of MDA-MB-231 apoptotic cell death by sodium butyrate using flow cytometric analyses.

To date, there are no studies assessing the induction of apoptosis by inosine in the MCF7 cancer cells. In the present study, inosine at 3 mg/mL (11.18 mM) demonstrated significantly more (*p* < 0.0001; Figure 2a,b) apoptotic cell death compared to the negative control, with 34.83% of the MCF7 cells in the early- and late-stage apoptosis after 24 h of treatment. The initiation of apoptotic cell death by inosine is indicative of its anticancer potential; however, further investigation is required to assess its impact on cell cycle, as well as the potential signalling pathways involved in this activity. Earlier studies reported that inosine exhibited cytoprotective effects on normal breast cells during hypoxic conditions in breast cancer; however, that study did not examine the potential anticancer properties of inosine against the MCF7 cells [20]. As such, the present study is one of the first to report on both the cytotoxicity and the induction of apoptosis by inosine in the MCF7 cells. Inosine was not previously investigated for its action against the MDA-MB-231 cells with no reports on its potential apoptotic activity. We observed that inosine initiated early- and late-stage apoptotic cell death by 25.82% after 24 h at 3 mg/mL (11.18 mM); however, no statistical difference was observed compared to the negative control (*p* > 0.05). A previous study discussed the molecular pathways involved in the induction of apoptosis by biologically similar compounds (adenosine and ATP) to inosine and reported the cytoprotective action of inosine in breast cancer hypoxic conditions [20]. As the current study only measures the activity of inosine against the MDA-MB-231 cells after 24 h of treatment, future studies can evaluate its impact at different time points. Furthermore, the action of inosine on the MDA-MB-231 cell cycle should also be explored.

Nisin at 3 mg/mL (894.44 μM) showed significantly more (*p* < 0.01; Figure 2a,b) combined early- and late-stage apoptotic cell death (25.73%) in the MCF7 cells compared to the negative control. This finding is not in line with the anti-proliferative activity findings for nisin in the present study. This discrepancy can be attributed to the differences in the time-points of the two assays. Notably, the anti-proliferative activity was measured after 72 h whilst the flow cytometry assessment was performed after 24 h. Flow cytometric assessment at different time points, including at 72 h, may provide further insights into the apoptotic potential of nisin. The current findings on the induction of apoptosis by nisin were in line with the existing studies [21,22]. Earlier, an in vitro study observed apoptotic cell death-related morphological changes in the MCF7 cells following treatment with nisin with no cell lysis [21]. However, that study only utilized visual observations with an optical microscope of the cells upon treatment [21]. Despite extensive preclinical research investigating the anticancer action of nisin against other cancer types, limited in vitro studies have assessed its activity against breast cancer. Notably, previous studies did not report the cytotoxic or apoptotic potential of nisin against the MDA-MB-231 cells. In the flow cytometric analyses, nisin induced combined early- and late-stage apoptosis in the MDA-MB-231 cells by 26.77% after 24 h at 3 mg/mL (894.44 μM), with no statistical difference observed compared to the negative control (*p* > 0.05). Interestingly, nisin was observed to initiate small levels of necrotic cell death (2.56%). A previous study acknowledged that calcium influx accompanied, but did not cause, neuronal necrosis [39], which could account for the induction of calcium influx and necrotic cell death by nisin in the MCF7 cells in the present study. However, no studies to date have investigated the effect of nisin on necrosis against breast cancer cells or other cancer types. Notably, necrosis was not observed with the other two gut microbial metabolites in either the MCF7 or the MDA-MB-231 cells in this study. However, the positive control, doxorubicin, displayed 40.12% necrosis in the present study, which was expected based on previous literature [40].

### 2.5. Proteomic Label-Free Bottom-up Quantification via Nano-UPLC-qTOF-MS

To quantify significant proteins involved in the anticancer action of the tested metabolites against the MCF7 cells, the raw proteomics data were filtered based on the following parameters: a *p*-value of at least ≤ 0.05, *q*-value ≤ 0.05, |log_2_fold change| ≥ 2, and unique peptide number of >0 were considered significant and included for further pathway analyses. The MCF7 breast cancer cell line was utilised, as most studies on gut microbial metabolites against breast cancer have been performed on this cell line. The resulting statistically significant proteins were further categorised based on their upregulation or downregulation by the metabolite treatments. To understand the molecular pathways involved based on the quantified proteins, several online software programs were implemented to specify the potential pathways triggered by the metabolites, including g:Profiler (Ensembl version 107, Ensembl Genomes version 54, Wormbase ParaSite version 17), Reactome (version 83), UniProt (release 2022_04), and STRING (version 12.0).

#### 2.5.1. Protein Identification and Quantification in the MCF7 Cells after the Treatment with Sodium Butyrate

Whilst studies have assessed the anticancer potential of sodium butyrate on the MCF7 cell line, the molecular mechanisms of action involved have not been studied extensively, with most studies utilizing fewer specific assays and technologies [11,14,30,34,35,41,42,43,44,45,46]. Reports have consistently indicated high levels of cytotoxicity by sodium butyrate at varying concentrations against the MCF7 breast cancer cell line [11,13,14,30,31,32,33,34,35,36,41], with several studies reaching similar conclusions regarding its molecular mechanisms of action. In the present study, sodium butyrate was determined to be the most effective gut microbial metabolite against the MCF7 and MDA-MB-231 breast cancer cell lines with induction of apoptosis and inhibition of ROS production. It also displayed less cytotoxicity toward the normal MCF10A human breast epithelial cells. In the proteomics analyses, sodium butyrate exhibited minimal modulation of cell cycle processes, in comparison to nisin and inosine; however, it significantly regulated the activity of signal transduction and metabolic processes in the MCF7 cells. Specifically, *PSAP* (prosaposin) demonstrated extensive potential activity both stand-alone and in conjunction with the expression of other genes upon sodium butyrate treatment. The upregulation of *PSAP* as a stand-alone effect potentially activated surfactant metabolism, and its combined upregulation with *GRB2* initiated a probable signal regulatory protein family interaction within the MCF7 cells. Additionally, the combined effect of *PSAP* with six other upregulated genes indicates a potential initiation of platelet activation, signalling, and aggregation. Further experimental data will be required to validate these findings. The upregulation of *PSAP*, *PYGB*, *CYB5R1*, *NEU1*, and *COTL1*, in combination with the downregulation of *VAMP8*, *PYCARD*, *IRAG2*, and *OSTF1*, indicated a potential initiation of neutrophil degranulation (Table 2). Of note, the combined upregulation of *GRB2* and *PTPN1*, coupled with the downregulation of *LIG1*, initiated a possible negative regulation of MET activity. The strong association between these genes can be seen in Figure 3. In addition, the combined upregulation of *GRB2* and *PTPRS* potentially initiated signalling by *NTRK3* (*TRKC*).

PSAP is a pleiotropic growth factor that has been identified to be upregulated in breast cancer cell lines; however, the precise role of this secreted protein in breast cancer is currently undetermined [47]. An in vivo study assessed the activity of this gene against the MCF7 and the T47D breast cancer cells in four-week-old female *v*/*v* mice, and the overexpression of PSAP was found to increase the ER α-mediated signalling axis and consequently to contribute to the development and progression of these specific breast cancer types [47]. That animal study observed that upregulation of PSAP increased the level of ERα protein expression dose-dependently in the tested cell lines and enhanced ERα-mediated transcription in an oestrogen-independent manner [47]. However, it was also determined that PSAP significantly encouraged tumour growth in the mice in comparison to the control treatment and potentially activated cell proliferation within the selected cell lines [47]. Interestingly, our study did not observe interactions between the *PSAP* gene and oestrogen receptor-related pathways; however, it did determine the action of *PSAP* in initiating multiple signalling pathways within the MCF7 cell line. Comparatively, an in vitro study identified that increased expression of the PSAP protein can inhibit the metastatic process via stimulation of p53 activity in MDA-MB-231 breast cancer cells, whereas inhibition of PSAP expression led to an increase in metastasis [48]. Whilst the findings of the two studies [47,48] differed in the action of PSAP in the context of breast cancer, both studies recognised that PSAP can serve as a biomarker in the malignancy signature of breast cancer, which could further be utilised as a therapeutic target for anticancer agents. Similarly, GRB2 (growth factor receptor-bound protein-2) was previously found to be markedly overexpressed in specific breast cancer cell lines, including the MCF7 cells, whereas it was significantly downregulated in normal breast epithelial tissue [49]. The authors identified that GRB2 activated the Ras signalling pathway, which regulated the growth factor sensitivity of human breast cancer cell lines [49]. It is well known that GRB2 is responsible for significant cellular functions, and its inhibition blocks cell proliferation and transformation of multiple cell types and impairs developmental processes in various organisms [50]. GRB2 also links the epidermal growth factor receptor tyrosine kinase to the activation of Ras signalling and ERK1 and ERK2, two Ras downstream kinases [50]. GRB2 requires further investigation into its effect on other kinases based on its significant role in signal transduction [50]. The recognition of the vital role of GRB2 in signal transduction processes is consistent with the findings of the present study demonstrating its role in initiating negative regulation of MET activity, signalling by *NTRK3*, cell-to-cell communication, platelet signalling and aggregation, and signalling of regulatory family protein interactions. A previous study assessed the effect of a *GRB2*-targeted therapeutic miR-27b-3p on the MDA-MB-231 and the MCF7 breast cancer cells to inactivate the MAPK/ERK and PI3K/AKT signalling pathways [51]. The *GRB2*-targeted therapeutic was observed to inhibit cell proliferation and colony formation in the tested breast cancer cell lines, including the MCF7, and was found to induce controlled apoptotic cell death and activated synergistic action with paclitaxel [51]. Notably, the GRB2-targeted therapeutic demonstrated potential in combatting multi-drug chemoresistance commonly seen with standard drugs for breast cancer [51]. The present study also observed that the combined activity of the upregulation of *KPNA2* and downregulation of *TFF1* potentially influenced oestrogen-dependent gene expression, which is a strong factor of consideration when examining the activity of sodium butyrate against oestrogen-dependent breast cancer types, including the MCF7. This probable combined activity of *KPNA2* and *TFF1* is consistent with the findings of a previous study that assessed the regulation of gene expression in HER-2-negative breast cancer patients, and observed the upregulation of *KPNA2* and downregulation of *TFF1* [52]. The oestrogen hormone can modulate gene expression via binding to oestrogen receptors (ERs), which bind to DNA and can alter DNA transcription [53]. This is a notable physiological function by the oestrogen hormone, and the modulation of oestrogen-dependent gene expression by sodium butyrate is a significant finding in the examination of this compound against the oestrogen-dependent MCF7 cells.

#### 2.5.2. Protein Identification and Quantification in the MCF7 Cells after the Treatment with Inosine

As there is no existing research on the activity of inosine against the MCF7 cells, the present study is the first to report on the molecular pathways activated in this cell type after inosine treatment. Overall, inosine regulated the expression of proteins involved in the modulation of cell cycle processes in the MCF7 cells. Similar to nisin, inosine potentially initiated the downregulation of the *PLK1* gene, which was involved in multiple cell cycle-related pathways, as depicted in Figure 4. As a stand-alone interaction, the downregulation of *PLK1* indicated a probable initiation of both polo-like kinase-mediated events and mitotic metaphase/anaphase transition. The combined interaction of the downregulation of *PLK1* and *SMC3* implied the activation of mitotic telophase/cytokinesis processes, the upregulation of *RRM2* and *TK1* and the downregulation of *PLK1* and *SMC3* potentially activated mitotic processes within the cell cycle of MCF7 cells after inosine treatment. Notably, the combined upregulation of *RRM2* and *TK1* indicates a potential initiation of G_1_/S-specific transcription, G_1_/S transition, and mitotic G_1_ phase and G_1_/S transition, which may imply that treatment with inosine induced cell cycle arrest in the G_1_/S growth phases. This could be confirmed with future cell cycle-specific assays. Furthermore, the increased expression of the P08243 protein, initiating expression of *ASNS*, in the MCF7 cells by inosine treatment could indicate the activation of multiple pathways, including *ATF4* activation of genes in response to endoplasmic reticulum stress, response of *EIF2AK1* (HRI) to heme deficiency, and PERK regulation. Significant interactions between genes following inosine treatment in the MCF7 cells are visually represented by the STRING network in Figure 4 and are also outlined in relation to their molecular activities in Table 2.

As aforementioned, upregulation of the *PLK1* gene was found to be associated with poor prognosis in breast cancer patients and may serve as a biomarker for diagnosis [54], whereas downregulation of this gene correlated with the induction of cell cycle arrest in the G_2_/M growth phase and inhibited the proliferation of the MCF7 breast cancer cells [55]. Our findings demonstrated that inosine potentially downregulated *PLK1*, indicating its potential interactions with the cell cycle processes that could contribute to apoptosis in the MCF7 cells. The upregulation of the *RRM2* (ribonucleotide reductase subunit 2) upon inosine treatment was another key observation against the MCF7 cells, as it potentially activated multiple cell cycle processes in combination with the increased gene expression of *TK1*, specifically in the G_1_/S growth phase of the cell cycle as indicated by the Reactome analyses. *RRM2* has been identified as a potential proliferation biomarker in different cancer types, including breast cancer, due to its high expression [56,57]. An in vitro study assessed the activity of an *RRM2*-targeted drug treatment against human cervical cancer cells, in which it was observed that the downregulation of *RRM2* correlated with the induction of apoptotic cell death and cell cycle arrest, and these findings were further supported by subsequent in vivo studies assessing the role of *RRM2* [57]. These findings conflict with the observations of our study, as we found an upregulation of *RRM2* in the MCF7 cells and the induction of apoptotic cell death upon inosine treatment. Another study assessed the activity of *RRM2* in an adrenocortical cancer cell line and reported that it had a greater upregulation in the S phase compared to the G_1_ phase, which caused a block in the S phase and an increase in cell abundance within the G_1_ phase [56]. The probable initiation of activity within the G_1_ and S cell cycle phases by the upregulation of the *RRM2* correlated with the observations of the present study. As with the previous studies on different cancer types, expression of *RRM2* was found to be increased in breast cancer patients in comparison to normal breast tissue [58,59]. The identification of *RRM2* as the malignancy signature indicates a potential therapeutic target for new anticancer therapies. A previous study also observed that downregulation of *RRM2* correlated with an inhibition in cell proliferation and growth in breast cancer cell lines, including triple negative and luminal subtypes, which demonstrated the potential premise of implementing *RRM2* inhibitors as therapeutic agents in cancer [58]. Upregulation of the *RRM2* was coupled with an upregulation of the *TK1* (Thymidine kinase 1), contributing to the potential modulation of the cell cycle molecular pathway. *TK1* has served as a biomarker of cell proliferation and has demonstrated promising prognostic value in early-stage breast cancer patients [60], and increased gene expression of *TK1* was associated with disease progression and poor treatment outcomes in specific breast cancer types [61], which indicated a potential use of this gene to monitor treatment outcomes in breast cancer patients and act as a therapeutic target. *TK1* regulates cell cycle and plays a significant role in DNA synthesis under normal physiological conditions; however, previous studies on its potential as a biomarker for breast cancer progression did not extensively report its cell cycle regulation in a tumour microenvironment. Whilst the utilisation of this gene as a prognostic breast cancer indicator has been explored, the molecular pathways associated with this gene are not well understood. Furthermore, *SMC3* was observed to be downregulated by the inosine treatment in the MCF7 cells in the present study. The impact of an *SMC3* inhibitor in the MCF7 cell line was also evaluated previously, and it was observed that *SMC3* downregulation induced apoptotic cell death and inhibited cell proliferation and cell cycle progression at G_1_ [62]. Whilst the previous studies on the role of *SMC3* in cohesin complex and other cohesin subunits are available in the literature [62], further investigation is warranted to understand its fundamental role in breast cancer.

#### 2.5.3. Protein Identification and Quantification in the MCF7 Cells after the Treatment with Nisin

As nisin induced apoptotic cell death in the MCF7 cells, the differential expression of key proteins in the MCF7 cells after 24 h of treatment with nisin was analysed using label-free LC-MS/MS proteomics. A comprehensive summary of the molecular action of nisin is depicted in Figure 5, indicating that nisin stimulated the dynein–dynactin complex and kinase-binding processes in the MCF7 cells. As molecular studies on the anticancer activity of nisin against the MCF7 cells is limited, the present study provided novel insights into the molecular pathways activated by nisin in this cell type. Nisin potentially triggered the activation of multiple components of the cell cycle in the MCF7 cells. It was found to upregulate the Q9HC77 and A0A7P0Z4C3 proteins, which could initiate the upregulation of the *DCTN1* (dynactin subunit 1) and *CENPJ* (centromeric protein J) genes, and downregulated P53350 protein, which instigated the downregulation of the polo-like kinase 1 (*PLK1*) gene. The collective upregulation and downregulation of these specific genes could be associated with the initiation of G_2_/M transition and mitotic G_2_-G_2_/M growth phases within the cell cycle, as well as mitotic metaphase/anaphase transition by the *PLK1* gene individually, as determined by the Reactome (version 83) software. The combined activity of the *DYNC1LI1*, *DCTN1*, *CENPJ*, and *PLK1* genes potentially further triggered mitotic prometaphase within the MCF7 cells. These observations indicate a strong effect on the cell cycle by nisin.

Under normal physiological conditions, the *CENPJ* gene is associated with the gamma–tubulin complex [63] and it maintains normal spindle morphology and centrosome integrity during cell division processes [64]. A study observed that the increased expression of *CENPJ* led to the induction of apoptosis and inhibition of cell proliferation in HeLa cells, in which cell cycle arrest was initiated in the G_2_/M phase [65]. These previous preclinical findings are of particular significance, as the present study also identified potential activity in the G_2_/M cell cycle growth phase following the overexpression of *CENPJ*, as well as observing the induction of MCF7 apoptotic cell death by nisin. In another preclinical study, the inhibition of apoptotic cell death was observed to be associated with the repression of caspase-3 by the *DCTN1* gene [66]. In an in vitro study assessing dynactin subunits in hepatocellular carcinoma (HCC), the *DCTN1* subunit was found to have no impact on the prognosis of HCC, despite this gene being upregulated in HCC patients in comparison to normal tissue, and the increased levels of the dynactin subunits correlated with the occurrence of AFP-negative expression in HCC patients [67]. Whilst that study did not provide a detailed overview of the *DCTN1* gene activity in relation to cancer, it acknowledged that the *DCTN2* subunit promoted cell cycle progression as an oncogene in HCC [67], which indicated that it promoted cell cycle in a cancer microenvironment and could serve as a therapeutic target. *DCTN1* was also identified to be upregulated in colon adenocarcinoma tissue in comparison to normal tissue [68]. Notably, a study by Wang et al. [68] also determined that gene expression of specific dynactin subunits, including *DCTN1*, was directly related to the cell cycle, apoptosis, and metastatic pathways, which correlated with the findings of the present study, indicating that the potential upregulation of this gene was linked with the initiation of multiple cell cycle processes. The increase in cell division processes and growth of the MCF7 cells could also account for the high cell viability values after 72 h of treatment with nisin observed in the Alamar Blue assay in the present study. Furthermore, nisin possibly downregulated the expression of the *PLK1* gene in the MCF7 cells. Previously, *PLK1* was reported to regulate cell cycle in the MCF7 cells and was upregulated during mitotic processes [55], which correlated with the findings of the present study on the role of this gene in potentially activating mitotic prometaphase, as well as metaphase/anaphase transition. The effect of *PLK1* overexpression on the cell cycle progression includes perturbations to cytokinesis, an increase in larger cells with fragmented nuclei, and delayed mitotic progression [69]. Interestingly, another study identified that *PLK1* offered therapeutic potential in oestrogen-receptor positive breast cancer cell lines, including the MCF7; however, the overexpression correlated with poorer prognosis in breast cancer patients, including the MDA-MB-231 cell line [54]. The impact of the *PLK1* inhibitor RO3280 on the cell proliferation and molecular pathways of breast cancer cell lines was also assessed earlier, and the MCF7 cell line was found to be the most sensitive to the treatment and produced the most promising results [55]. The inhibitor was observed to inhibit cell proliferation, induced cell cycle arrest in the G_2_/M phase and apoptotic cell death, caused DNA damage, and also reduced the mitochondrial membrane potential in the MCF7 cells [55]. These findings are in line with the present study, as the nisin treatment acted as a potential *PLK1* inhibitor against the MCF7 cells and potentially initiated regulation of *PLK1* activity at the G_2_/M transition of the cell cycle. Based on the present findings and previous preclinical research, more in vitro and in vivo studies are necessary to elucidate the potential therapeutic role of nisin in breast cancer.

## 3. Materials and Methods

### 3.1. Preparation of Gut Microbial Metabolite

The gut microbial metabolites nisin, inosine, and sodium butyrate and positive standard doxorubicin (DOX) were purchased from Sigma Aldrich, Castle Hill, NSW, Australia. Stock concentrations of 6.5 mg/mL (1.94 mM), 15 mg/mL (55.92 mM), and 500 mg/mL (4.54 M) for nisin, inosine, and sodium butyrate, respectively, were prepared in sterile Milli-Q water based on their solubility. A stock concentration of 1 mM DOX was prepared in sterile DMSO.

### 3.2. Determination of Cancer Cell Viability against MCF7 and MDA-MB-231 Cells, and Potential Cytotoxicity on the MCF10A Human Breast Epithelial Cells

The MCF7 (RRID:CVCL_0031) human breast adenocarcinoma, MDA-MB-231 (RRID:CVCL_0062) human breast adenocarcinoma and the MCF10A (RRID:CVCL_0598) human breast epithelial cells were purchased from the American Type Culture Collection (ATCC, Manassas, VA, USA) and cultured and maintained in the PC2 Pharmacology Laboratory at the NICM Health Research Institute (NICM HRI), Western Sydney University as per the previously described method [70]. Briefly, the MCF7 and MDA-MB-231 cells were grown in DMEM (Dulbecco’s Modified Eagle Medium; Lonza Australia, Mount Waverley, VIC, Australia) comprised of 4.5 g L^−1^ glucose, L-glutamine, and sodium pyruvate supplemented with 10% fetal bovine serum (FBS; In Vitro Technologies, Lane Cove West, NSW, Australia) and 1% penicillin-streptomycin (Sigma-Aldrich, NSW, Australia). The MCF10A cells were grown in MEGM (Mammary Epithelial Cell Growth Medium; Lonza Australia, Mount Waverley, VIC, Australia) comprised of 2.00 mL BPE, 0.50 mL hEGF, 0.50 mL insulin, 0.50 mL hydrocortisone, and 0.50 mL GA-1000 (Lonza Australia, Mount Waverley, VIC, Australia). Normal FBS-containing DMEM and MEGM media were used as the negative controls for the breast cancer and the normal breast cells, respectively, whereas 1 μM of DOX was used as the positive control. In a 96-well plate, 100 μL of MCF7, MDA-MB-231, and MCF10A was seeded in each well at a cell count of 1 × 10^5^/mL and incubated overnight at 37 °C in the presence of 5% CO_2_ to adhere to the base of the well. These cells were then treated with different concentrations of sodium butyrate, nisin and inosine (in triplicate) for 72 h, ranging from 0.03125–3 mg/mL. To determine the potential anti-proliferative activity of the gut metabolites against the breast cancer and normal breast epithelial cells, the study utilised the Alamar Blue assay at the end of the 72-h period [70]. This assay measures the level of oxidation during cellular respiration which is directly proportional to the number of viable cells [70]. Following the 72-h incubation period, the media was removed from the wells and 100 μL of the Alamar Blue dye (0.1 mg/mL) was added to each well and incubated for 3 h at 37 °C in the presence of 5% CO_2_. The fluorescence was measured using a microplate spectrophotometer (BMG CLARIOstar, Mornington, VIC, Australia) at an emission wavelength at 595 nm and an excitation wavelength at 555 nm. The cell viability of the cell lines was calculated as a percentage of the negative control (DMEM for breast cancer cells and MEGM normal breast epithelial cells).

### 3.3. Reactive Oxygen Species (ROS) Assay

Elevated levels of ROS contribute to carcinogenesis and tumour metastasis [40], which illustrates the significance of ROS detection in the proposed study. The intracellular ROS level in the MCF7 and MDA-MB-231 breast cancer cells after treatment with sodium butyrate, nisin and inosine was evaluated using the DCFDA (2′,7′-dichlorofluorescein diacetate) Cellular ROS Detection Assay Kit (#ab113851; Abcam, Melbourne, VIC, Australia) according to the manufacturer’s protocol and as per the method described by Alsherbiny, Bhuyan [40]. In a 96-well plate, 100 μL of MCF7 and MDA-MB-231 cells was seeded in each well at a cell count of 2.5 × 10^5^/mL and incubated overnight at 37 °C in the presence of 5% CO_2_ to adhere to the base of the well. The MCF7 and MDA-MB-231 cells were stained with DCFDA fluorogenic dye and incubated for 45 min at 37 °C in the presence of 5% CO_2_. Following incubation, the wells were washed with 1× buffer and treated with the two concentrations (2 and 3 mg/mL) of the three gut metabolites. DOX (1 μM) was used as the positive drug control, tert-butyl hydrogen peroxide (TBHP; 250 μM) as the control for ROS production, and supplementary buffer (provided with the kit) as the negative control. The cells were then incubated for 4 h at 37 °C in the presence of 5% CO_2_. The fluorescence was measured using a microplate spectrophotometer (BMG CLARIOstar, Mornington, VIC, Australia) at an emission wavelength at 535 nm and an excitation wavelength at 485 nm.

### 3.4. Flow Cytometric Analysis Evaluation of the Apoptotic Profiles of the Cells

Based on the cell viability data, flow cytometry was used for the evaluation of apoptosis in the MCF7 and MDA-MB-231 cells after treatment with sodium butyrate, nisin and inosine as per the protocols described earlier using an annexin V and 7-AAD-based kit (#ab214663, Abcam, Melbourne, VIC, Australia) [40,71]. Annexin V detects phosphatidylserine on the external membrane of apoptotic cells, whereas 7-AAD is a DNA fluorescent and dead-cell marker that indicates any disruption of cell membrane structural integrity [71]. Apoptosis is an early-occurring molecular event, and most studies determined the induction of apoptosis within 24 h to accurately detect both early- and late-stage apoptotic death [40,72].

The MCF7 and MDA-MB-231 cells were cultured in T75 cell culture flasks using DMEM with 10% FBS at a seeding density of 1 × 10^5^/mL and were incubated overnight at 37 °C in the presence of 5% CO_2_ to adhere to the back wall of the flask. The media was aspirated and replaced with fresh FBS-containing media, and the cultured flasks were then treated with the highest concentration of each gut metabolite (3 mg/mL), DOX at a concentration of 1 μM as the positive control and non-FBS-containing media as the negative control and incubated for 24 h. Following the 24-h incubation period, the suspension cell media contained in each flask was collected. The adherent cells remaining in each flask were then treated with 0.25% *w*/*v* of trypsin (TrypLE™ Express Enzyme, Thermo Fisher Scientific, North Ryde, NSW, Australia) for 5 min, neutralized and combined with the suspension cell media. The cell media was then centrifuged at 500× *g* for 5 min at room temperature (RT) to collect the cell pellet. The pellet was resuspended in 1 mL of PBS and centrifuged at 500× *g* for 5 min, and this step was then repeated. Harvested cell pellets were resuspended in 0.5 mL of 1× binding buffer at a final cell concentration of 1 × 10^6^ cells/mL, and 5 μL of annexin V-CF blue and 7-AAD staining solutions were added to each 200 μL of cell suspension. The cells were then incubated at 37 °C in the presence of 5% CO_2_ in darkness at RT for 15 min, after which 400 μL of 1× binding buffer was added. Then, 200 μL of the new suspension was added to a 96-well V-bottom plate in triplicate for each treatment. The cells were then analysed by ACEA Biosciences Novocyte 3000 flow cytometer (ACEA Biosciences Inc., San Diego, CA, USA).

The NovoExpress (version 1.5.0, ACEA Biosciences Inc., San Diego, CA, USA) software was implemented for analysis and processing. Furthermore, cells were gated on FSC vs. SSC to exclude the debris near the origin and cell aggregates. To determine which cells were viable and which were undergoing apoptotic cell death, dot-plots of annexin V-CF in Pacific Blue vs. 7-AAD fluorescence in PerCP were gated in a quadrant, in which the lower-left quadrant indicated live cells (+Annexin V and −7-AAD), the lower-right quadrant early apoptotic cells (+Annexin V and −7-AAD), the upper-right quadrant late apoptotic cells (+Annexin V and +7-AAD), and the upper-left quadrant necrotic cells (−Annexin V and +7-AAD).

### 3.5. Proteomics Analyses

#### 3.5.1. Cell Culture, Treatment, and Protein Extraction

The MCF7 cells were cultured in T75 cell culture flasks using DMEM with 10% FBS at a seeding density of 1 × 10^5^/mL and were incubated overnight at 37 °C in the presence of 5% CO_2_ to adhere to the back wall of the flask. The media was aspirated and replaced with fresh DMEM with 10% FBS, and the cultured flasks were then treated with the highest concentration of each gut metabolite (3 mg/mL), and DMEM (no FBS) as the negative control and incubated for 24 h at 37 °C in the presence of 5% CO_2_. To gain a more comprehensive understanding of the potential molecular pathways activated, 3 mg/mL was chosen for the proteomics analyses, as all three metabolites presented with the greatest level of cytotoxicity at this concentration. In alignment with prior studies, the present proteomics study was completed after a 24 h treatment period to adequately capture all apoptosis-related protein changes. Following the 24-h incubation period, the suspension cell media contained in each flask was collected. The adherent cells remaining in each flask were then treated with 0.25% *w*/*v* of trypsin (TrypLE™ Express Enzyme, Thermo Fisher Scientific, North Ryde, NSW, Australia) for 5 min, neutralized and combined with the suspension cell media. The cell media was then centrifuged at 500× *g* for 5 min at RT to collect the cell pellet. The pellet was resuspended in 1 mL of PBS and centrifuged at 500× *g* for 5 min, and this step was then repeated. The cell pellets were then resuspended in 100 µL of lysis buffer (#1863441, Thermo Fisher Scientific, North Ryde, NSW, Australia) including 1 µL of universal nuclease and fully mass spectrometry (MS)-compatible Halt™ Protease and Phosphatase Inhibitor Cocktail, EDTA-Free (Thermo Fisher Scientific, North Ryde, NSW, Australia). The cells were pipetted up and down 10–15 times until the sample viscosity was reduced and the cell pellets were left on ice for 20 min with occasional vertexing for 10 s every 5 min, then centrifuged at 14,000 rpm for 20 min at 4 °C, and the lysate was collected.

#### 3.5.2. Protein Quantification

Following lysate collection, Pierce^TM^ Rapid Gold BCA Protein Assay Kit (#A53226, Thermo Fisher Scientific, North Ryde, NSW, Australia) was used to determine the protein concentration of the cell lysate in triplicate against bovine serum albumin (BSA) standard according to the manufacturer’s protocol. Briefly, 1 μL of each sample replicate (*n* = 3) was diluted in the MilliQ water (at 1:20) with 20 μL of each standard and placed in a 96-well plate with 200 μL of working reagent per well. Samples were diluted to be within the operating range of 20–2000 μg/mL. The plate was mixed thoroughly on a plate shaker for 30 s and incubated at RT for 5 min, and then the absorbance was measured within 20 min at 480 nm using a microplate spectrophotometer (BMG CLARIOstar, Mornington, VIC, Australia). The blank absorbance was subtracted from all other readings of standards and samples, and sample concentration was determined against the established BSA standard calibration curve. Protein samples were stored at −80 °C for future use and further proteomic analysis.

#### 3.5.3. Peptide Preparation and Clean up

The EasyPep^TM^ Mini MS Sample Prep Kit was utilised as per the manufacturer’s protocol (Thermo Fisher Scientific, North Ryde, NSW, Australia) for chemical and enzymatic sample processing, using 100 μg of protein samples. Using lysis buffer (#1863441, Thermo Fisher Scientific, North Ryde, NSW, Australia), the final volume was adjusted to 100 μL in a microcentrifuge tube. The reduction and alkylation solutions were added at a volume of 50 μL each, gently mixed, and further incubated at 95 °C for 10 min using a heat block. The samples were cooled at RT, and 50 μL of the reconstituted trypsin/lys-C protease mixture was added to each sample, then incubated (with shaking) at 37 °C for 3 h. Following incubation, 50 μL of digestion stop solution was added and gently mixed. Peptide clean-up columns were implemented to remove both hydrophobic and hydrophilic contaminants. Following the clean-up, clean peptide samples were then dried using a vacuum centrifuge and resuspended in 100 μL 0.1% formic acid in water for the LCMS analysis, and gently placed in maximum recovery sample vials (Waters Corp., Milford, MA, USA).

#### 3.5.4. Label-Free Bottom-Up Quantification via Nano-Ultra High-Performance Liquid Chromatography Coupled with Quadruple Time of Flight Mass Spectrometry (Nano-UPLC-qTOF-MS)

The LC-MS data for the present study were acquired according to the outline methodologies of two previous studies [40,72]. Briefly, a nanoACQUITY UPLC system (Waters Corp., Milford, MA, USA) coupled to a Synapt G2-S high-definition mass spectrometer (HDMS) (Waters Corp., Milford, MA, USA), operating in positive electron spray ion mode (ESI+) and equipped with a hybrid quadrupole time of flight (qTOF) analyser, was used to analyse tryptic peptides. Waters NanoLockSpray Exact Mass Ionisation Source was utilised to maintain mass accuracy. Further, 100 fg/mL Glu-fibrinopeptide B (GFP) dissolved in 50% aqueous acetonitrile containing 0.1% formic acid with a lock mass *m/z* of 758.84.26 was infused as lockspray solution at 0.5 μL/min and calibrated against a sodium iodide solution. For the chromatographic separation of peptides, the nanoEase M/Z BEH C18 (1.7 μM, 130 Å, 75 μm × 100 mm, Waters Corp., Milford, MA, USA) at 40 °C was utilised and coupled to a nanoEase M/Z Symmetry C18 Trap Column (100 Å, 5 µm, 180 µm × 20 mm, (Waters Corp., Milford, MA, USA). Milli-Q water and acetonitrile containing 0.1% formic acid were used as mobile phase A and B, respectively (LC-MS grade; Merck, Macquarie Park, NSW, Australia). An injection volume of 1 μL at 300 nL/min flow rate was used throughout the 50-min gradient. Samples were injected into the trapping column at 5 μL/min at 99% mobile phase A for 3 min before being eluted to the analytical column. An initial 1% of mobile phase B was ramped to 85% B over 50 min with the following gradient: 10% B at 2 min, 40% B at 40 min, and 85% B at 42 min. All samples were kept at 4 °C and were injected in triplicate. The ion source block temperature was set to 80 °C and capillary voltage was maintained at 3 kV. Ions were acquired with *m/z* between 50 and 2000, scanning time of 0.5 s, sample cone voltage and source offset at 30 V, nanoflow gas at 0.3 Bar, purge gas at 20 L/h and cone gas flow at 20 L/h. The data-independent acquisition (DIA) method by MS^E^ multiplex mode was used for sample acquisition at a T-wave collision-induced dissociation cell filled with argon gas with MassLynx Mass Spectrometry Software (version 4.2, Waters Corp., Milford, MA, USA).

#### 3.5.5. Data Processing and Availability

Progenesis QI software (version 2.0, Waters Corp., Milford, MA, USA) was used to import and further process the acquired MassLynx data. Automatic selection of alignment reference among QC samples was set, and peptides were identified against the Uniprot human proteome database (May 2022 version) using the ion-accounting method with 250 kDa protein mass maximum. One fragment per peptide or one peptide per protein together with 3 fragments per protein were set as the ion-matching requirements using relative quantification implementing the Hi-N method (*n* = 3). Auto peptide and fragment tolerance and less than 4% FDR were set for search tolerance parameters. Peptides with absolute mass error >20 ppm or single-charged were further filtered out. Pairwise comparisons of the identified proteins in the treated groups were carried out against the control group for cytotoxic potential exploration. In each experimental design, proteins with analysis of variance (ANOVA)-derived *p*-value at least ≤ 0.05 and q value ≤ 0.05 with |log_2_fold change| ≥ 2 were considered significant and included for further pathway analyses. Differentially expressed proteins identified by the quantitative processing of the LC-MS/MS analysis of the proteome tryptic digestion were analysed by STRING (https://string-db.org/, accessed on 14 November 2022) [73], g:Profiler (https://biit.cs.ut.ee/gprofiler/gost, accessed on 14 November 2022) [74], Reactome (https://reactome.org/, accessed on 14 November 2022) [75], and IMPaLA (http://impala.molgen.mpg.de/, accessed on 14 November 2022) [76] to identify the relevant pathways responsible for the action against the MCF7 breast cancer cells. The g:SCS algorithm was used for multiple-testing corrections in the g:Profiler platform with an adjusted *p* value of 0.05 threshold. The raw and processed data were further deposited to the ProteomeXchange Consortium via the PRoteomics IDEntifications (PRIDE) repository [77] with the dataset identifier 10.6019/PXD040482.

#### 3.5.6. Statistical Analyses

The GraphPad Prism (version 9, La Jolla, CA, USA) was used to determine the IC_50_ (concentration that inhibits cell growth by 50%) values, as well as to compare the mean values between the control and treatment groups of the Alamar Blue, ROS and flow cytometry experiments. This included performing the independent samples *t*-test and one and two-way ANOVA. Data were further expressed as the mean ± SD. Statistical differences between the mean values in the experiments of at least *p* < 0.05 were considered statistically significant for the purpose of the study.

## 4. Conclusions

In recent years, preclinical research has been conducted to understand the anticancer potential of gut microbial metabolites, the by-products of residing gut microbiota. Specific gut microbial metabolites including SCFAs, bacteriocins, and natural purine nucleosides (such as inosine), have demonstrated potential anticancer activity against different cancer types including breast cancer. As a common SCFA, sodium butyrate has been one of the most extensively investigated gut metabolites against breast cancer, including the two studied cell lines of the present study, MCF7 and MDA-MB-231 adenocarcinoma cells. The present study observed that sodium butyrate inhibited the proliferation of both breast cancer cell lines with a maximum inhibitory percentage of 80% at 27.25 mM, which correlated with the previous preclinical studies on this metabolite. Nisin, one of the well-known bacteriocins, has also been investigated previously for its anticancer potential, but primarily against colorectal cancer cell lines. The findings of this study determined that nisin exhibited minimal cytotoxicity against breast cancer cells, which was in contrast with previous studies that observed high cytotoxicity by nisin against the MCF7 adenocarcinoma cells. Inosine has been minimally investigated for its anticancer potential with no studies against breast cancer cells. Inosine at 11.18 mM was found to demonstrate cell inhibition of 47.78% and 48.85% in the MCF7 and MDA-MB-231 cells, respectively. All three gut metabolites were also found to initiate cytotoxicity in normal MCF10A human breast epithelial cells. However, sodium butyrate induced lower cytotoxicity in the MCF10A cells than in the MCF7 and MDA-MB-231 adenocarcinoma cells, which warrants further investigation in animal models. All three selected gut metabolites were found to significantly inhibit the production of ROS in both breast cancer cell lines, which is a promising aspect, as an increase in ROS production can be associated with an increased risk of cancer development and metastasis. Additionally, sodium butyrate was observed to initiate apoptotic cell death in both cell lines. As apoptosis is a desired hallmark feature for prospective anticancer agents, our findings provided meaningful insights for future in vivo and clinical studies on sodium butyrate. In the present study, proteomics analyses were also performed for the MCF7 adenocarcinoma cells, and sodium butyrate was found to potentially activate several signal transduction pathways underpinning the observed anti-proliferative activity. Comparatively, nisin and inosine indicated the potential activation of cell cycle processes in the MCF7 cells. Given the promising findings of the present study, future studies must implement cell viability, flow cytometric and proteomics assays to understand the impact of different timepoints and doses of these metabolites against breast cancer cells. Additionally, future research should prioritize the use of 3D cell culture such as spheroids, representing a more accurate preclinical model to assess the activity of potential anticancer drugs in comparison to 2D cell culture. Collectively, the present study provides a strong foundation for future preclinical research to ascertain effective therapeutic doses and administration routes of the selected gut microbial metabolites, as well as how these metabolites interact with the gut microbiota, the tumour microenvironment, and the immune system.

## Figures and Tables

**Figure 1 ijms-24-15053-f001:**
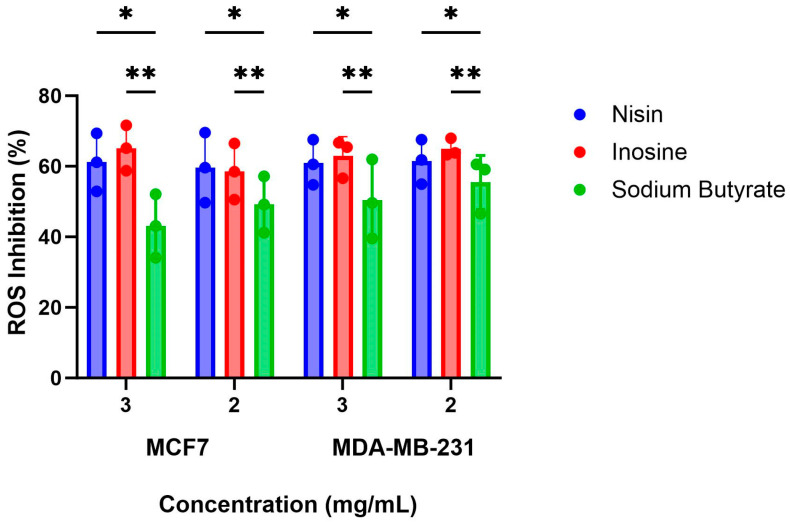
Reactive oxygen species (ROS) inhibitory percentage for nisin, inosine, and sodium butyrate, at 3 and 2 mg/mL, against the MCF7 and MDAMB231 cells after 4 h of treatment (*n* = 3). * Indicates *p*-value: 0.01 < *p*-value < 0.05. ** Indicates *p*-value ≤ 0.01. For the MCF7 cells, inosine exhibited significantly more ROS inhibition than did sodium butyrate at 3 mg/mL (*p* < 0.05). For the MDA-MB-231 cells, inosine, at both 3 and 2 mg/mL showed significantly more ROS inhibition (*p* < 0.05) compared to sodium butyrate at 3 mg/mL.

**Figure 2 ijms-24-15053-f002:**
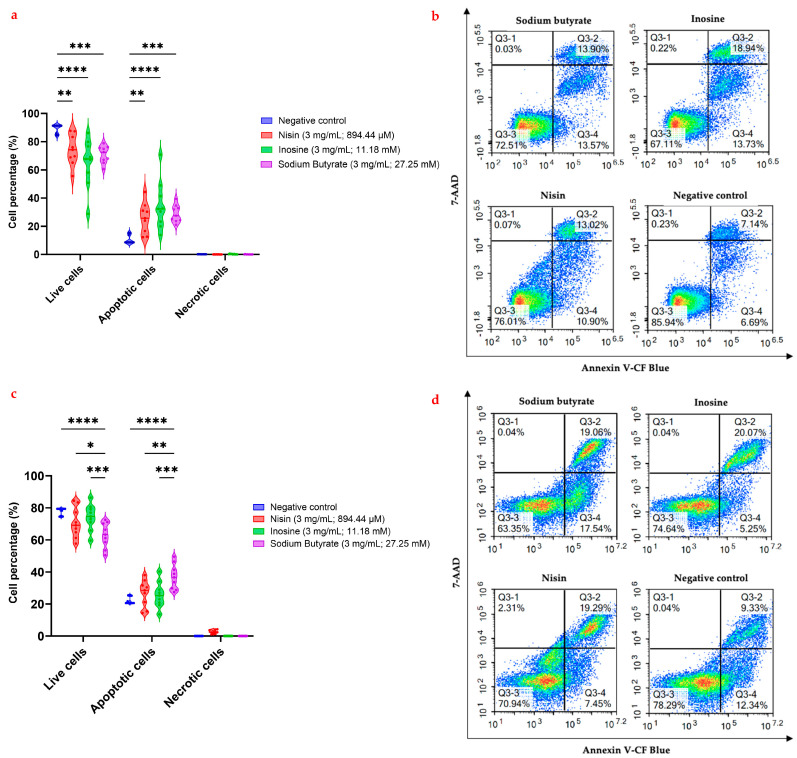
Flow cytometric assessment of the apoptotic profiles of the MCF7 (**a**,**b**) and MDAMB231 (**c**,**d**) breast cancer cells after 24 h of treatment with nisin, inosine, and sodium butyrate (*n* = 3). * indicates 0.01 < *p*-value < 0.05; ** indicates *p* ≤ 0.01; *** indicates *p* ≤ 0.001; **** indicates *p* ≤ 0.0001 compared to the negative control. (**a**) All three metabolites showed significantly lower live cells and higher apoptotic (early- and late-stage) cells compared to the negative control in the MCF7 cell line. Notably, inosine exhibited the greatest activity among the three metabolites with significantly lower live cells (*p* < 0.0001) and higher apoptotic (early- and late-stage) cells (*p* < 0.0001) compared to the negative control. (**b**) Represented are the density plots of each drug treatment that is most representative of the average data from the flow cytometric analyses, with Q3-1 = necrotic cells, Q3-2 = late-stage apoptotic cells, Q3-3 = live cells, and Q3-4 = early-stage apoptotic cells. (**c**) All three metabolites showed significantly lower live cells and higher apoptotic (early- and late-stage) cells compared to the negative control in the MDAMB231 cell line. Sodium butyrate exhibited significantly lower live cells (*p* < 0.05) and greater apoptotic (early- and late-stage) cells (*p* < 0.01) compared to nisin, inosine and the negative control. (**d**) The density plots of each drug treatment shown here are the most representative of the average data from the flow cytometric analyses, with Q3-1 = necrotic cells, Q3-2 = late-stage apoptotic cells, Q3-3 = live cells, and Q3-4 = early-stage apoptotic cells.

**Figure 3 ijms-24-15053-f003:**
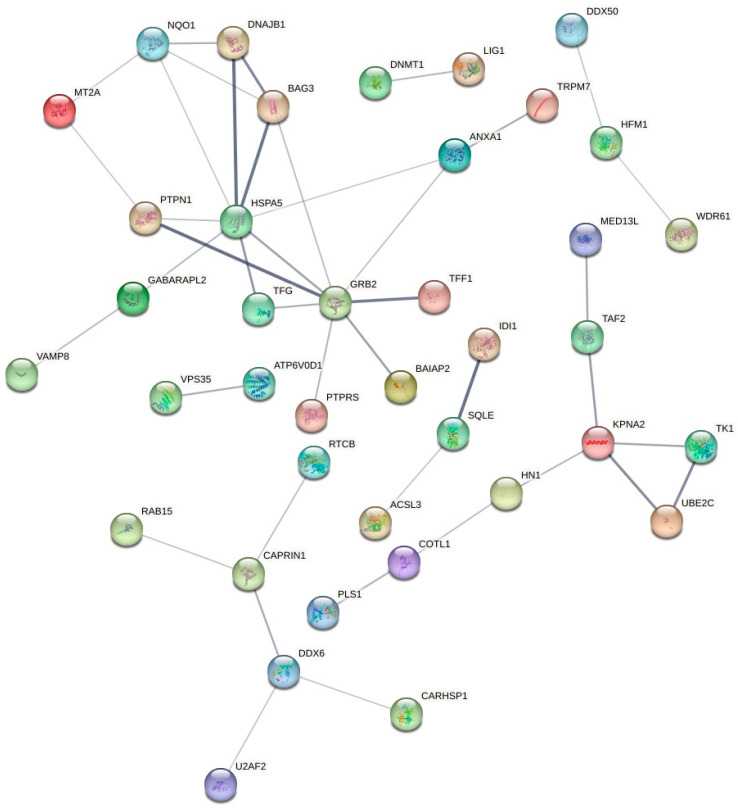
The STRING network of associated expressed genes in the MCF7 cells following sodium butyrate treatment for 24 h, with the connecting lines depicting significant interactions between multiple genes. The thicker lines indicate a more significant association, whereas the thinner lines are indicative of a less significant association.

**Figure 4 ijms-24-15053-f004:**
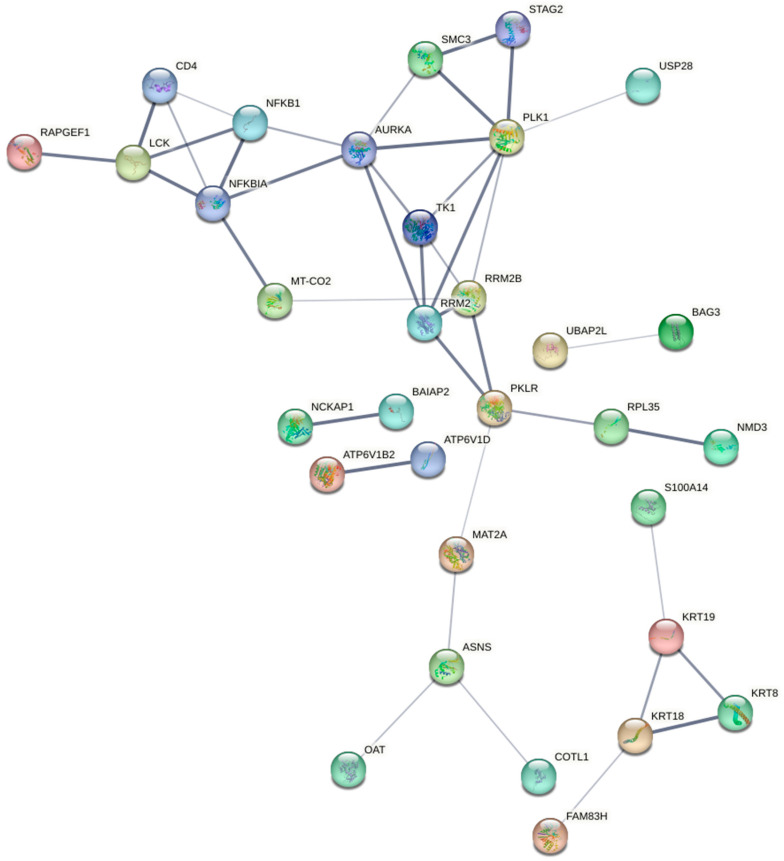
The STRING network of associated expressed genes in the MCF7 cells following 24 h of inosine treatment, with the connecting lines depicting significant interactions between multiple genes. The thicker lines indicate a more significant association, whereas the thinner lines are indicative of a less significant association.

**Figure 5 ijms-24-15053-f005:**
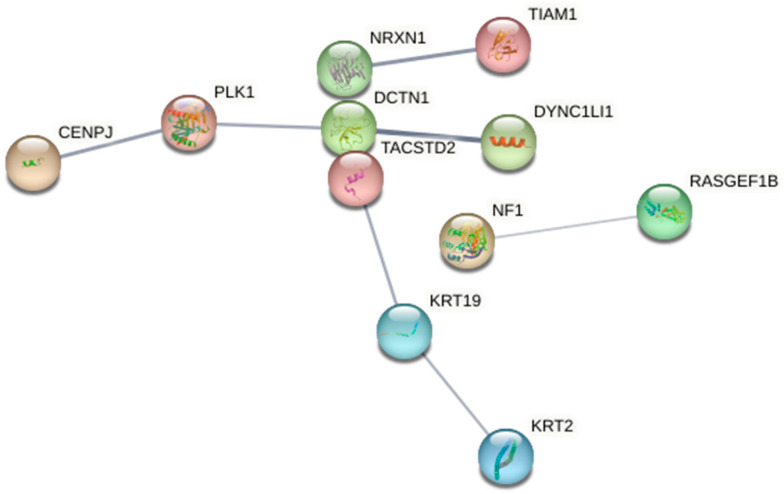
The STRING network of associated expressed genes in the MCF7 cells following 24 h of nisin treatment, with the connecting lines depicting significant interactions between multiple genes. The thicker lines indicate a more significant association, whereas the thinner lines are indicative of a less significant association.

**Table 1 ijms-24-15053-t001:** Cell growth inhibition (%) of nisin, inosine, and sodium butyrate against the MCF7 and MDA-MB-231 breast adenocarcinoma and the MCF10A normal breast epithelial cell lines at concentrations of 0.03125–3 mg/mL for 72 h using the Alamar Blue assay (*n* = 3).

Concentration (mg/mL)	Cell Growth Inhibition (%)								
	MCF7			MDA-MB-231			MCF10A		
	Sodium Butyrate	Inosine	Nisin	Sodium Butyrate	Inosine	Nisin	Sodium Butyrate	Inosine	Nisin
3	81.73 ± 4.81 ^a^	47.78 ± 8.89 ^b^	14.34 ± 15.66 ^c^	80.80 ± 1.06 ^a^	48.85 ± 1.11 ^b^	24.46 ± 6.79 ^c^	89.23 ± 0.92 ^a^	87.64 ± 5.51 ^a^	69.43 ± 10.28 ^a^
2	70.86 ± 7.24 ^a^	39.88 ± 7.04 ^b^	13.31 ± 7.43 ^c^	67.70 ± 6.28 ^a^	36.09 ± 3.32 ^b^	12.29 ± 7.97 ^c^	83.96 ± 4.65 ^a^	84.47 ± 7.38 ^a^	54.76 ± 15.69 ^a^
1	58.15 ± 2.90 ^a^	37.25 ± 5.07 ^b^	13.97 ± 9.09 ^c^	71.44 ± 6.84 ^a^	27.80 ± 6.40 ^b^	12.01 ± 4.13 ^c^	66.53 ± 3.85 ^a^	69.49 ± 13.77 ^a^	34.57 ± 15.22 ^b^
0.5	43.84 ± 3.49 ^a^	19.12 ± 5.39 ^b^	8.86 ± 5.47 ^b^	51.05 ± 2.71 ^a^	13.67 ± 3.55 ^b^	10.18 ± 5.13 ^b^	39.46 ± 12.27 ^a^	31.29 ± 18.71 ^a^	27.20 ± 12.77 ^a^
0.25	35.79 ± 3.44 ^a^	4.04 ± 5.61 ^b^	6.92 ± 6.47 ^b^	31.33 ± 3.83 ^a^	6.38 ± 1.67 ^b^	6.95 ± 5.39 ^b^	18.47 ± 17.41 ^a^	20.94 ± 14.73 ^a^	2.80 ± 29.91 ^a^
0.125	25.98 ± 4.62 ^a^	-	5.89 ± 4.10 ^b^	18.45 ± 5.87 ^a^	4.01 ± 0.75 ^b^	6.33 ± 7.06 ^b^	8.39 ± 18.78 ^a^	12.11 ± 19.81 ^a^	18.27 ± 13.50 ^a^
0.0625	17.62 ± 3.88 ^a^	-	2.48 ± 5.71 ^b^	7.57 ± 5.49 ^a^	2.08 ± 0.41 ^a^	2.41 ± 4.27 ^a^	3.76 ± 7.58 ^a^	18.71 ± 15.44 ^a^	21.05 ± 10.93 ^a^
0.03125	10.78 ± 2.08 ^a^	-	0.98 ± 7.78 ^a^	4.93 ± 5.23 ^a^	3.52 ± 0.92 ^a^	3.90 ± 1.87 ^a^	5.67 ± 8.31 ^a^	3.57 ± 17.48 ^a^	15.98 ± 6.31 ^a^
IC_50_	0.576 mg/mL (5.23 mM)	2.743 mg/mL (10.23 mM)	ND	0.557 mg/mL (5.06 mM)	3.262 mg/mL (12.16 mM)	ND	0.655 mg/mL (5.95 mM)	0.655 mg/mL (2.44 mM)	1.63 mg/mL (485.98 μM)

^a,b,c^ The different subscript values in the same row for each cell line indicate statistically significant difference (*p* < 0.05) between the treatment groups. “-” indicates no cell inhibition. “ND” indicates not detected. Data are presented as mean ± standard deviation (SD).

**Table 2 ijms-24-15053-t002:** The upregulation (red ↑) and downregulation (blue ↓) of the most prolific proteins and genes (*p* < 0.02) by the gut microbial metabolites in the MCF7 breast cancer cell line, including the associated molecular pathways and mechanisms of action induced by the expression of these proteins and genes.

Gut Metabolite	Regulation	Protein(s)	Gene(s)	Molecular Pathway	Mechanism of Action
Sodium butyrate	↑	C9JIZ6	*PSAP*	Metabolism of proteins	Surfactant metabolism
				Disease	Defective *CSF2RA* causes SMDP4
					Defective *CSF2RB* causes SMDP5
					Diseases associated with surfactant metabolism
					Defective *SFTPA2* causes IPF
	↑	E7EVQ6 Q13907	*SQLE* *IDI1*	Metabolism	Activation of gene expression by SREBF (SREBP)
					Regulation of cholesterol biosynthesis by SREBP (SREBF)
	↑	C9JIZ6 P62993	*PSAP* *GRB2*	Cell-cell communication	Signal regulatory protein family interactions
	↑	P02795 P21291	*MT2A* *CSRP1*	Cellular responses to stimuli	Response to metal ions
	↑ ↓	P18031 P62993 F5GZ28	*PTPN1* *GRB2* *LIG1*	Signal transduction	Negative regulation of MET activity
	↑	P21291	*CSRP1*	Cellular responses to stimuli	*MTF1* activates gene expression
	↑	P18031 Q9UHQ9 P11021 C9JIZ6 P62993 E9PI30 P04899	*PTPN1* *CYB5R1* *HSPA5* *PSAP* *GRB2* *CTSW* *GNAI2*	Hemostasis	Platelet activation, signalling and aggregation
	↑ ↓	P52292 P04155	*KPNA2* *TFF1*	Signal transduction	Oestrogen-dependent gene expression
	↑	P11021	*HSPA5*	Cellular responses to stimuli	*ATF6* (*ATF6*-alpha) activates chaperone genes
					*ATF6* (*ATF6*-alpha) activates chaperones
	↑ ↓	P11216 Q9UHQ9 C9JIZ6 Q99519 Q14019 Q9BV40 Q9ULZ3 Q12912 Q92882	*PYGB* *CYB5R1* *PSAP* *NEU1* *COTL1* *VAMP8* *PYCARD* *IRAG2* *OSTF1*	Immune system	Neutrophil degranulation
	↑ ↓	P25685 P11021 O95817	*DNAJB1* *HSPA5* *BAG3*	Cellular responses to stimuli	Regulation of *HSF1*-mediated heat shock response, cellular response to heat stress
	↑ ↓	P52292 P04899 P04155	*KPNA2* *GNAI2* *TFF1*	Signal transduction	ESR-mediated signalling
	↑ ↓	P25685 P02795 P11498 P11021 P21291 F5GYQ1 O00762 O95817 P15559	*DNAJB1* *MT2A* *PC* *HSPA5* *CSRP1* *ATP6V0D1* *UBE2C* *BAG3* *NQO1*	Cellular responses to stimuli	Cellular responses to stimuli
	↑	P11216	*PYGB*	Metabolism	Glycogen breakdown (glycogenolysis)
	↑	C9JIZ6 P62993 Q15404	*PSAP* *GRB2* *RSU1*	Cell-cell communication	Cell-cell communication
	↑	Q13332 P62993	*PTPRS* *GRB2*	Signal transduction	Signalling by *NTRK3* (*TRKC*)
	↑	Q9UHQ9 P11021 C9JIZ6 E9PI30	*CYB5R1* *HSPA5* *PSAP* *CTSW*	Hemostasis	Platelet degranulation
					Response to elevated platelet cytosolic Ca^2+^
Inosine	↑	P31350 P04183	*RRM2* *TK1*	Cell cycle	G_1_/S Transition and G_1_/S-Specific Transcription
					Mitotic G_1_ phase and G_1_/S transition
	↑	P08243	*ASNS*	Cellular responses to stimuli	Response of *EIF2AK1* (HRI) to heme deficiency
					*ATF4* activates genes in response to endoplasmic reticulum stress
					PERK regulates gene expression
	↑	P08727 P05783 P05787	*KRT19* *KRT18* *KRT8*	Developmental biology	Formation of the cornified envelope
					Keratinisation
	↓	P53350 Q9UQE7	*PLK1* *SMC3*	Cell cycle	Mitotic Telephase/Cytokinesis
	↑ ↓	P21281 P08243 F2Z388	*ATP6V1B2* *ASNS* *RPL35*	Cellular responses to stimuli	Cellular response to starvation
	↑	Q9Y2A7 H0Y6R4 I3L4C2	*NCKAP1* *RAPGEF1* *BAIAP2*	Signal transduction	*RAC3* GTPase cycle
	↑	P30613	*PKLR*	Developmental biology	Regulation of gene expression in beta cells
	↑ ↓	P08243 F2Z388	*ASNS* *RPL35*	Cellular responses to stimuli	Response of *EIF2AK4* (*GCN2*) to amino acid deficiency
	↑	Q9Y2A7 I3L4C2	*NCKAP1* *BAIAP2*	Signal transduction	*RHO* GTPases activate WASPs and WAVEs
	↑ ↓	P31350 P04183 Q9UQE7 P53350	*RRM2* *TK1* *SMC3* *PLK1*	Cell cycle	Cell Cycle, Mitotic
	↑	P51572 Q9Y2A7 P06239 H0Y6R4 Q5VT25 I3L4C2	*BCAP31* *NCKAP1* *LCK* *RAPGEF1* *CDC42BPA* *BAIAP2*	Signal transduction	*RHO* GTPase cycle
	↑	P31350	*RRM2*	Gene expression (transcription)	Transcriptional regulation by *E2F6*
	↑	H0YAV1 P00403	*RRM2B* *MT-CO2*		*TP53* Regulates metabolic genes
	↓	P53350	*PLK1*	Cell cycle	Polo-like kinase-mediated events
					Mitotic metaphase/anaphase transition
Nisin	↑ ↓	Q9HC77 A0A7P0Z4C3 P53350	*CENPJ* *DCTN1* *PLK1*	Cell cycle	Loss of proteins required for interphase microtubule organization from the centrosome
					Loss of Nlp from mitotic centrosomes
					*AURKA* activation by *TPX2*
					Recruitment of mitotic centrosome proteins and complexes
					Centrosome maturation
					Regulation of *PLK1* activity at G_2_/M transition
					Recruitment of NuMA to mitotic centrosomes
				Organelle biogenesis and maintenance	Anchoring of the basal body to the plasma membrane
				Cell cycle	G_2_/M transition
					Mitotic G_2_-G_2_/M phases
	↓	P53350	*PLK1*	Cell cycle	Polo-like kinase-mediated events
					Mitotic metaphase/anaphase transition
	↑ ↓	Q9Y6G9 A0A7P0Z4C3 Q9HC77 P53350	*DYNC1LI1* *DCTN1* *CENPJ* *PLK1*	Cell cycle	Mitotic prometaphase

## Data Availability

The raw and processed data were deposited to the ProteomeXchange Consortium via the PRoteomics IDEntifications (PRIDE) repository [77] with the dataset identifier 10.6019/PXD040482. The raw data is also shared in the Appendix A.

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
