# Peer review of "Mechanistic Insights into the Anti-Proliferative Action of Gut Microbial Metabolites against Breast Adenocarcinoma Cells"

_ijms, 2023, doi:10.3390/ijms242015053_

Round 1

Reviewer 1 Report

The manuscript prepared by Kayla Jaye and coauthors, aims to study the biological effect of selected

gut microbiota metabolites (sodium butyrate, inosine, and nisin) on MCF7 and MDA-MB-231 human breast cancer cell lines. The text is very interesting for the readers, since recently there is growing evidence about the effect of postbiotics on human body health. The manuscript as a text is really well prepared – the methodology is presented with almost all needed details allowing the repeat of selected experiment, the results are also presented very clearly (i.e., this is in vitro study and there is even presentation of used chemicals concentrations in different units). However, after reading the text I have some experimental doubts and questions to Authors which need to be answered before manuscript publication, and  even some experiments need to be performed. If possible, the answers and comments should be also included in a new version of manuscript, since they will allow better understanding of presented data.

Since the Authors performed some experiments on MCF10A cell line, please explain its usage shortly in the Introduction part.

In Methods I do not see the reason why the cytotoxicity for MCF10A cells is presented in a separate chapter, since the method is the same as for MCF7 and MDA-MB-231 cell lines.

The most important thing I would like to know is what was the rationale for cells incubation with compounds for 72 h (cell viability studies), whereas the proteomics and apoptosis analysis were performed after cells treatment with compounds for 24 h (and with the highest concentration).

3.6.1 – why only 3 mg/mL concentration was used in the study?

Line 749 – confirm that cells were left to adhere “to the back wall of the flask”(what type of flask (25, 75 cm2?), not to the main surface, and if so - than why?

Line 759 – what lysis buffer was used?

Figure 1 – on Y-axis instead of “cells inhibition” I suggest to use “cytotoxicity”; add SD and n; change the concentration presented on the X-axis (from smallest to the highest). On the other hand to make the results more clear I suggest to present data for MCF10A as another columns in Table 1.

In Figures and Tables add information about incubation time, n of experiments.

3.4- why ROS level was determined after cells incubation for only 4 h and only for the highest concentrations of compounds – 2 and 3 mg/mL? What is the chance that in vivo the cells can be treated with such high concentrations of compounds? Despite this I suggest to perform the experiment showing the effect of all studied compounds at all concentration after cells incubation for 72 h.

I do not see the coherence with the ROS determination after 4 h, apoptosis determination and proteomics studies after 24 h, and the cytotoxic effect determination after 72 h.

Figure 2 – on Y-axis present rather “ROS level”; there is no bar for DOX, no bar for control cells.

2.4 – additionally present the data showing the effect of compounds on metabolic activity after 24h incubation, as well as the ROS level. Have the Authors performed similar studies with normal MCFA10 cells?

Since it is in vitro cell line based study I suggest to use “cytotoxicity” instead “toxicity” in the text.

What is the physiological concentration of selected chemicals (postbiotics) in media after selected strains of bacteria culture? Are theses concentrations correlated with the studied? What about the metabolism and bioavailability of studied compounds? Can the authors propose/list the bacteria strains (as artificial “formulation”) that can be potentially active producers of studied metabolites? Have the Authors studied the effect of mixture of compounds prepared in a similar way but with the  quantities of each component determined in media after bacteria growth? Maybe even the culture medium (as a real source of postbiotics) have been studied?

In summary, the manuscript requires at the least the mayor revision before its publication.

Author Response

Please see the attachment. Please also note that the response letter includes the responses to all three reviewers' feedback - note for this reviewer, only refer to the addressed comments under Reviewer 1 in this document. 

Reviewer 2 Report

In this research article, the authors investigated the effect of gut-microbiota-derived metabolites on breast cancer cell lines and normal breast epithelial cells. The major concern in this study is that the compounds show non-specific cytotoxic activity. As shown in Figure 1, both 2000 and 3000 μg/mL are cytotoxic to normal cell lines, exhibiting comparable inhibition to that against the cancer cell lines. Furthermore, these concentrations were used to measure apoptotic cell death and ROS production in the cancer cell lines, while results from the normal cells are not included in the manuscript. Thus, no solid conclusions can be drawn on the cytotoxic capacity of these compounds and the translational value of the findings. The inclusion of another normal cell line in this design (especially those derived from the gut- as they would be the first in contact with compounds produced by the resident microbiota) could possibly provide clearer results.

Minor points:

Line 30: cell cycle processes? No experimental data on cell cycle progression are included in the manuscript.

Line 108: “cell inhibition” is not correct phrasing. It should be changed to anti-proliferative activity or decrease in cell viability etc.

Line 119: both Alamar blue and MTT assays measure cell viability indirectly by determining cellular metabolism. Why is that MTT is outdated and not Alamar blue. Additionally, to the best of my knowledge Western blot is not used to measure viability of cells.

Line 348-349: this discrepancy could be controlled by the authors. Why were those two timepoints selected?

Figure 3: the legend should include which graphs refer to what cell line.

Line 397: this is not a thesis.

Line 422: no experimental data for “...did initiate platelet activation, signaling, and aggregation.” are included in the manuscript. This is purely an assumption based on the proteomic data.

Lines 494, 499-502: no experimental data for: “…the downregulation of PLK1 led to the initiation of both polo-like kinase mediated events and mitotic metaphase/anaphase transition.” and for “the combined upregulation of RRM2 and TK1 initiated G1/S-specific transcription, G1/S transition, and mitotic G1 phase and G1/S transition, which may imply that treatment with inosine induced cell cycle arrest in the G1/S growth phases.” are included in the manuscript. These conclusions are purely an assumption based on the proteomic data.

Author Response

Please see the attachment. Please also note that the response letter includes the responses to all three reviewers' feedback - note for this reviewer, only refer to the addressed comments under Reviewer 2 in this document. 

Reviewer 3 Report

In the submitted manuscript authors analyzed impact of three gut microbiome postbiotics on the biology, ROS production and proteome composition of two breast cancer cell lines, and found that sodium butyrate was the most effective in inhibiting cell proliferation, ROS production, and initiating apoptotic cell death.

This manuscript is robust and well written, however there are some both major and minor drawbacks which must be corrected and further improved before this manuscript is suitable for publication:

1) Authors should re-check that all abbreviations used in the main text, like SCFA, GIT and DOX, were explained after their first mentioning.

2) Line 119: I'm really not aware that earlier studies have primarily used WB for assessment of cell viability, at least not in the last 20 years!

3) Authors should re-check that all tables and figures were properly numbered and cited in the text, since Table 2 is missing and Figure 4 was not cited in the text. Also, since Figure 3 has 4 panels, proper subpanel should be cited in the text where Figure 3 was mentioned.

4) For Table 1, it should be mentioned how data were presented (I assume, mean±SD), while proper text in line 193 should be "statistically significant difference".

5) It is unclear why impact of postbiotics on cancer cell lines was presented with a table, and on normal breast cell with a graph, which BTW neither have measure of dispersion nor results of statistical analysis. Those should be uniformly presented.

6) In the main text actual p-values should be provided (p=...), while levels of significance (asterisks, p<0.00.., p>0.05) only in figures. This particularly relates to Figure 2 legend, only where actual p-values were presented, while it was not explained what asterisks mean. Also, in Figure 2 and line 263, the proper name of cell line MDA-MB-231 should be stated.

7) Line 289: There is no need to capitalize the first letter in 15-lipoxygenase-1.

8) Text in lines 396-397, particularly "based on the nature of the thesis", is awkward and unclear.

9) Lines 393 and 831: It is unclear why statistical parameters in those two sentences differ, and since in Table 3 you showed both up- and down-regulated proteins, fold change should be put under two vertical straight lines which represent that an absolute value was meant!

10) Line 834: For all used on-line based tools, except its reference, a valid URL should be provided. Also, in line 836 the proper name is "g:SCS" algorithm.

11) Sentence in line 849-850 seems redundant since it is unclear were are results obtained with MetaboAnalyst 5.0, while in line 820 it was written that the name of software is "Progenesis QI".

12) For all used cell lines, the proper Resource Identification Initiative ID from https://www.cellosaurus.org/ should be cited in sections 3.2 and 3.3, like for MCF-7 (RRID:CVCL_0031).

13) Presentation of proteomic results is the weakest part of this manuscript! Since authors haven't performed transcriptomic analysis, like RNA-seq, they definitively cannot claim something like "Similar to nisin, inosine initiated the downregulation of the PLK1 gene"! Since correlation between proteome and transcriptome is far from 1.0, you can just speculate about postbiotics' impact on gene expression, but since you have evidence for their impact on protein expression, you should stick just to proteins, and don't mention genes! The same is with figure legends.

14) Line 903: Statement on data availability "The data are shared in the supplementary file." is meaningless, since the same thing mentioned in lines 837-839 should be stated here, while in both places precise description of Supplementary Table 1 should be provided, and those Excel file provided as supplementary should be adequately transformed into Supplementary Table 1.

Author Response

Please see the attachment. Please also note that the response letter includes the responses to all three reviewers' feedback - note for this reviewer, only refer to the addressed comments under Reviewer 3 in this document. 

Round 2

Reviewer 1 Report

I have read the answers; the Authors answered the most of my comments and they improved the manuscript. I would suggest to present the concentration of butyrate, nisin, inosine in mg/mL instead of microg/mL.

Aftera all, I suggest the acceptance of the manuscript for publication in IJMS.

Author Response

Please see the attachment. Please note that this reviewer will be referring to the comments under Reviewer 1 in the response to reviewers' letter. 

Reviewer 2 Report

I thank the authors for addressing most of my concerns. However, the issue of cytotoxicity observed in normal MCF10A breast epithelial cell lines is not adequately addressed by the authors. As I mentioned in my first report the authors should at least assess ROS inhibition and the apoptotic profile of normal MCF10A breast cells after treatment with nisin, inosine, and sodium butyrate at 3000 μg/mL and compare their findings to MCF7 and MDAMB231 cancer cells.

Author Response

Please see the attachment. Please note that this reviewer will be referring to the comments listed under Reviewer 2 in the response to reviewers' letter. 

Reviewer 3 Report

Authors have substantially improved quality of this manuscript during revision, and satisfactorily responded to all reviewers' concerns.

I just have two suggestions:

1) For including both up- and down-regulated proteins, the proper notation of required fold change is: |log2fold change|≥1

2) Although very catchy, the new title seems more appropriate for a review paper than for original research article, since it is not very informative about the content of this manuscript.

Author Response

Please see the attachment. Please note that this reviewer will be referring to the comments listed under Reviewer 3 in the response to reviewers' letter. 

Round 3

Reviewer 2 Report

Although there are still some issues as mentioned in my previous report, the resubmitted manuscript can be accepted for publication.